# Global analysis of temporal clusters of storm surges

Ariadna Martín[1,2] 🔾, Robert Jane[1,2], Alejandra R. Enriquez[3] and Thomas Wahl[1,2]

[1]Department of Civil, Environmental and Construction Engineering, University of Central Florida, Orlando, FL, USA; [2]National Center for Integrated Coastal Research, University of Central Florida, Orlando, FL, USA and [3]School of Geosciences, College of Arts & Sciences, University of South Florida, St Petersburg, FL, USA

## Research Article

**Keywords:**
storm surge; temporal clustering; Poisson distribution; inter-arrival time

**Corresponding author:**
Ariadna Martín;
Email: ariadna.martinoliva@ucf.edu

## Abstract

Temporal storm surge clustering refers to a series of events affecting the same region within a short period of time, which can strongly influence coastal flooding impacts and erosion. Here, we analyze global storm surge clustering from tide gauges and a state-of-the-art global model hindcast to identify geographical hotspots of extreme storm surge clusters and assess event frequencies. We study the spatial distribution as well as the contribution of different event intensities to clustering. On average, globally, 92% of coastal locations show significant temporal clustering for 1-year return period events, and 25% for 5-year return level events, although notable spatial differences exist. Our results reveal two distinct clustering regimes: (i) short timescale clustering, where events occur in rapid succession (intra-annual), and (ii) long timescales (inter-annual), providing varying recovery times between events. We also test the validity of assuming a Poisson distribution, commonly used in storm surge frequency analyses. Our results show that >80% of the stations analyzed do not follow a Poisson distribution, at least when including events that are not the most extreme but exceeded, for example, the 1-year return level. These findings offer insights into temporal clustering dynamics of storm surges and their implications for coastal hazard assessments.

## Impact Statement

Understanding when and where coastal storm surges tend to cluster over time is essential for improving disaster preparedness and risk management. This study identifies global hotspots where multiple extreme storm surge events can occur within short timeframes, increasing the risk of compounded impacts on communities, infrastructure and ecosystems. The findings challenge the traditional assumption that such events happen randomly and independently, showing instead that certain regions face higher risks due to clustering. These insights can help stakeholders by incorporating temporal clustering into coastal risk assessments, which can lead to more effective and resilient coastal management strategies.

## Introduction

Coastal flooding is among the most devastating natural hazards, causing substantial economic losses and human casualties worldwide. While such floods often result from the combination of multiple processes (e.g., tides, rainfall, wind-wave effects and storm surge), high storm surges are a key contributing factor. Here, we refer to storm surges as events where coastal sea levels are substantially higher than predicted as a response to changes in the mean sea-level pressure and wind-driven setup. Reliable estimates of the occurrences of these extreme events are essential for risk assessment, insurance companies and coastal communities. An emerging area of concern is the temporal clustering of apparently independent extreme storm surge events (i.e., when multiple events occur in quick succession rather than being randomly spaced over time). Temporal clustering may have important consequences for coastal infrastructure and ecosystems as recovery times between events are minimized, and cumulative effects compound the risk, especially when communities are still recovering from previous shocks (Zscheischler et al., 2020).

In September 2017, Hurricanes Irma and Maria impacted Puerto Rico within 2 weeks, causing severe winds, rainfall and storm surges. Similarly, the United Kingdom experienced a series of storms during the winter of 2013/2014, with an average recurrence of 2.5 days (Jenkins et al., 2022). More recently, in 2024, Hurricanes Helene and Milton impacted the Florida Gulf Coast in quick succession, mirroring previous cases such as Ian and Nicole (2023) and Charley and Jeanne (2004), where subsequent storms exacerbated damage in already affected areas. Figure 1 shows how the season of 2005 affected the US Gulf Coast with hurricanes and tropical cyclones and their associated storm surges. While not all clustered events are extreme, moderate storms can still cause significant damage, particularly when flood defenses and coastal protections are already

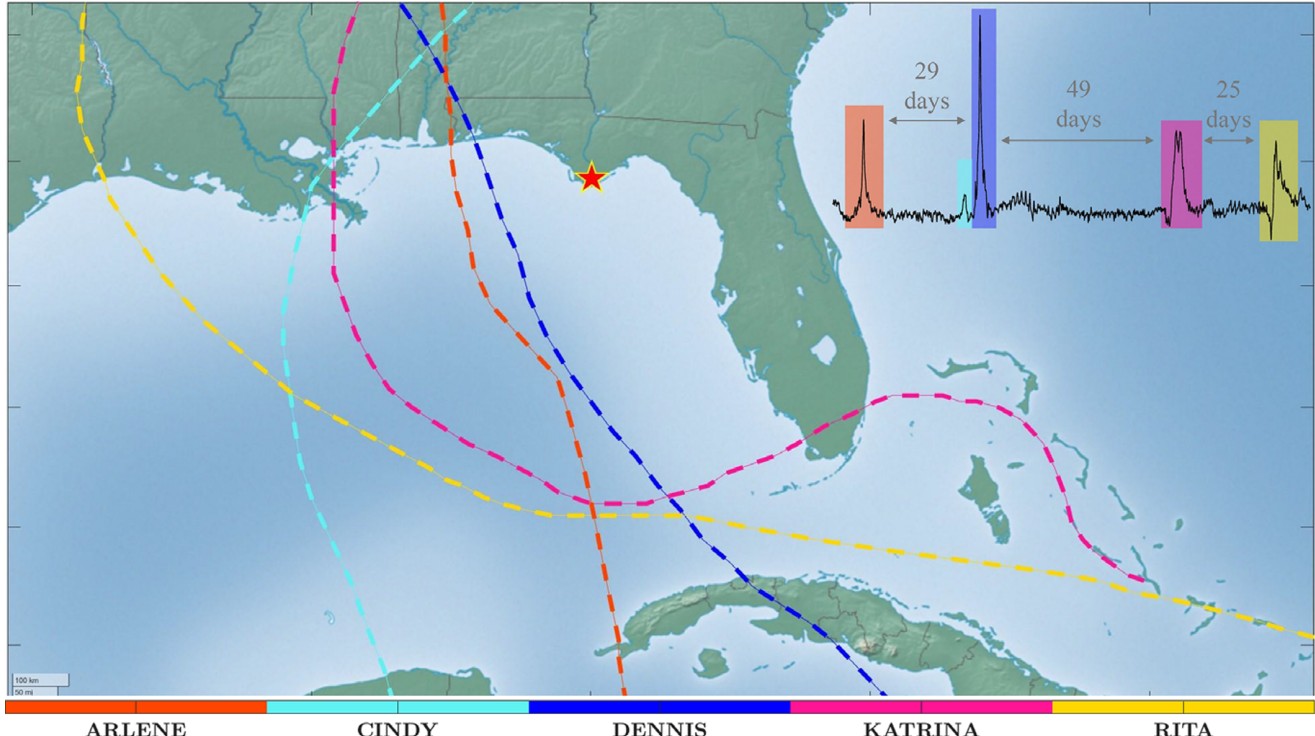

**Figure 1.** Tropical storms and hurricanes that produced storm surges at the Apalachicola River tide gauge (red star), Florida, in 2005. The tracks of the tropical storms and hurricanes are shown in dashed lines with different colors. The inset plot indicates the inter-arrival times between these different storm surge events.

weakened. Karunarathna et al. (2014) found that beach erosion from clusters of moderate storms can resemble the erosion caused by a single extreme event. Similarly, Macamo et al. (2016) observed that consecutive events can slow down the recovery process of mangrove forests, exacerbating the natural process of erosion and sedimentation in those areas.

In some cases, consecutive events can have less detrimental long-term effects. For example, when tropical cyclones occur within a season and are not scattered throughout the year. This creates an extended calm period, giving coral reefs and other natural ecosystems more time to recover than they would under a more random distribution of events with the same rate of occurrence (Mumby et al., 2011; Jagger and Elsner, 2012). This underscores the need to accurately predict both short-term clustering and longer recovery periods between storm clusters.

Despite extensive research on individual storm surge events, there is limited understanding of the temporal clustering patterns and their consequences. This recurrence of extreme events illustrates long-term problems with the miscommunication of risk to the public and decision-makers (Towe et al., 2020). Traditionally, event frequency has been modeled using the Poisson distribution (Jagger and Elsner, 2012), assuming independent and identically distributed events where event inter-arrival times are exponentially distributed. For example, the National Oceanic and Atmospheric Administration (NOAA) Experimental Long Lead Seasonal Hybrid Hurricane Forecast System, which uses a Poisson regression model based on Villarini et al. (2010). However, historically observed clustering, including those already discussed, suggests a higher probability of shorter inter-arrival times than expected under an exponential distribution. This could lead to a systematic under-prediction of both the number of events per season and the frequency of inactive seasons (Jagger and Elsner, 2012). Storm surge events impact a wide range of natural ecosystems, including beaches, dunes, as well as public infrastructure and coastal

communities. The recovery time of these systems varies significantly, ranging from months to decades, and is highly site-specific (Dollar and Tribble, 1993; Morton et al., 1994; Hamideh et al., 2018; de Ruiter et al., 2020). Back-to-back events can slow down these recovery times by exacerbating the impacts on these areas; therefore, the definition of a cluster is highly dependent on the system as well as the area of study and can substantially influence the results of the analysis.

Therefore, in this article, we present a comprehensive global analysis of storm surge clustering using a wide range of clustering definitions. We test the validity of the Poisson assumption under those definitions and assess how event intensity influences clustering patterns. Additionally, we analyze inter-arrival times, distinguishing between short-term clustering and calm periods (recovery time). The results provide critical insights into the temporal structure of storm surge events, contributing to improved risk assessment and coastal adaptation strategies.

The article is organized as follows: the second section describes the datasets and methodology. In the third section, we present the results, including model validation against tide gauge observations ("The role of clustering definition and validation of model hindcast data" section), the application of the methodology at the global scale ("Global analysis" section) and an analysis of inter-arrival times ("Inter-arrival time analysis" section). The findings are discussed in the fourth section, and conclusions are provided in the fifth section.

## Data and methods

### Data

To achieve a consistent spatial–temporal resolution of coastal sea level at the global scale, we use model-based sea-level data from the Coastal Dataset for the Evaluation of Climate Impact 2020 (CoDEC) (Muis et al., 2020, available at the Copernicus Climate

Change Service [2023]). Time series of sea level and tides are computed using the Deltares Global Tide and Surge Model (GTSMv3.0), a hydrodynamic model that dynamically simulates water levels at 10-min intervals using forcing fields from the ERA5 global reanalysis (Hersbach et al., 2020) from 1979 to 2018. The dataset includes 18,719 grid points along a smoothed global coastline, with a spatial resolution of 10–50 km. The nontidal residual time series (henceforth referred to as storm surge) are derived by subtracting the tides from the sea-level data.

We assess the model's ability to represent temporal clustering using in situ observations from the Global Extreme Sea Level Analysis database–version 3 (GESLA-3) (Woodworth et al., 2016; Haigh et al., 2022). A total of 527 coastal tide gauges from GESLA are used, those with at least 20 years of records between 1979 and 2018. To ensure consistency, gaps in the tide gauge storm surge data are mirrored in the CoDEC time series, and the mean sea-level variation is removed using a 30-day moving window, which leaves the seasonality due to atmospheric conditions (also captured by the hydrodynamic model used to produce the CoDEC data). The astronomical tides are calculated using the MATLAB U-Tide package (Codiga, 2025) on a year-by-year basis, for years with more than 70% of data availability. U-tide is applied using the least-squares method, white noise floor assumption for confidence intervals and an automated choice of constituents, resulting in 67 tidal constituents per year on average. Seasonal tidal components (i.e., solar annual (SA) and solar semi-annual (SSA)) were excluded from the harmonic analysis to preserve seasonality, which is not already removed by the 30-day moving average. The tidal analysis was performed on 369-day periods (12.5 lunar months) to consider a period close to multiples of the synodic periods of the short-term tidal constituents.

The independent storm surge events are identified following a declustering process set out in Martín et al. (2024). This method identifies a site-specific standard event duration, which is used to isolate independent extreme events. The method accounts for the temporal and spatial variability of the storms. To decluster the time series, a threshold is required. By varying this threshold, we evaluate the sensitivity of our results to different levels of extremeness. Specifically, we used five return levels (1–5 years), determined by fitting independent events to a generalized Pareto distribution.

## Methods

Usually, independent events are considered part of the same cluster if their inter-arrival time is shorter than a predefined time window. We refer to this time window as the "clustering window" hereinafter. To assess the sensitivity of results to different clustering window definitions, we use 14 different windows ranging from 15 days to 2 years, allowing us to explore the temporal structure of clustering within this period (exact values are: 15, 30, 45, 60, 75, 90, 120, 150, 180, 240, 300, 365, 545 and 730 days).

Another important factor is the severity of those consecutive events. Return levels are commonly used to quantify risk over a given period. For example, a 1-year return level represents an event expected to occur (or be exceeded) annually on average. However, clustering remains possible, leading to years with multiple exceedances above the 1-year return level threshold and others with none. To align with traditional risk threshold definitions, we use return levels to represent different intensity thresholds. Specifically, we apply five thresholds, corresponding to 1- to 5-year return levels,

to determine the number of events included in the analysis (see Supplementary Figure S1).

Stations with significant clustering violate the Poisson assumption, indicating storm surge clustering that cannot be explained by a Poisson process. In contrast, stations with no significant clustering may still exhibit clustering patterns, but consistent with a Poisson distribution. We test whether storm surges follow a Poisson-distributed event frequency by using *Ripley's K* metric (Ripley, 1981), which indicates the tendency for clustering within a time series. It is defined as the average number of events within a time window (i.e., our clustering window) around another event in the same time series. The higher the value, the more clustering occurs in that time series. We calculate this metric for all tide gauge stations and CoDEC grid points and assess statistical significance using bootstrap simulations, similar to the approach by Brunner and Stahl (2023). Specifically, we generate 1,000 homogeneous Poisson-distributed binary time series with the same rate of occurrence as the observed time series. Values of the *Ripley's K* index, for the observed time series, above the 98th percentile of the index calculated for the 1,000 Poisson-distributed series are deemed to indicate statistically significant clustering.

## Results

### *The role of clustering: Definition and validation of model hindcast data*

We start by testing the sensitivity of clustering against different clustering definitions and levels of extremeness, as well as comparing results from model hindcasts to those derived from observations. Our results indicate that extreme storm surges cluster in time at most sites, for both tide gauge observations and modeled data. The level of clustering, however, varies with the threshold. Figure 2 compares the percentage of time series exhibiting temporal clustering (Figure 2a,b) with the percentage of time series where the clustering is deemed significant, that is, the clustering behavior deviates from the Poisson distribution (Figure 2c,d). The analysis is performed using observations from GESLA (Figure 2a,c) and CoDEC model points (Figure 2b,d). Validation includes 527 tide gauges and their corresponding CoDEC points, using 14 clustering window definitions and 5 different thresholds.

Figure 2a,b show the percentage of stations (out of the 527 selected) that exhibit clustering at least once in the time series. More stations lack clusters at higher thresholds (going from 1 to 5-year return levels), which is to be expected due to fewer events exceeding these thresholds. As clustering windows lengthen, more events cluster, which is also to be expected, but the relationship is nonlinear. Particularly, for higher thresholds, less clustering occurs when using 200 days as the clustering window. When focusing on stations with significant clustering (Figure 2c,d), lower thresholds yield more significant clustering due to increased event counts. The number of stations with significant clustering decreases with increasing thresholds, but never falls below 20%, that is, at least 20% of stations exhibit significant clustering at any threshold. Peaks occur around 90- to 120-day windows, declining until 1-year windows, which indicates that there are certain windows when significant clustering is more likely.

The differences between general occurrences of clustering (i.e., where some clustering was observed) and occurrences of significant clustering (i.e., clustering is statistically significantly different compared to a Poisson distribution) (Figure 2a–d, respectively) indicate the number of stations where clustering happens but

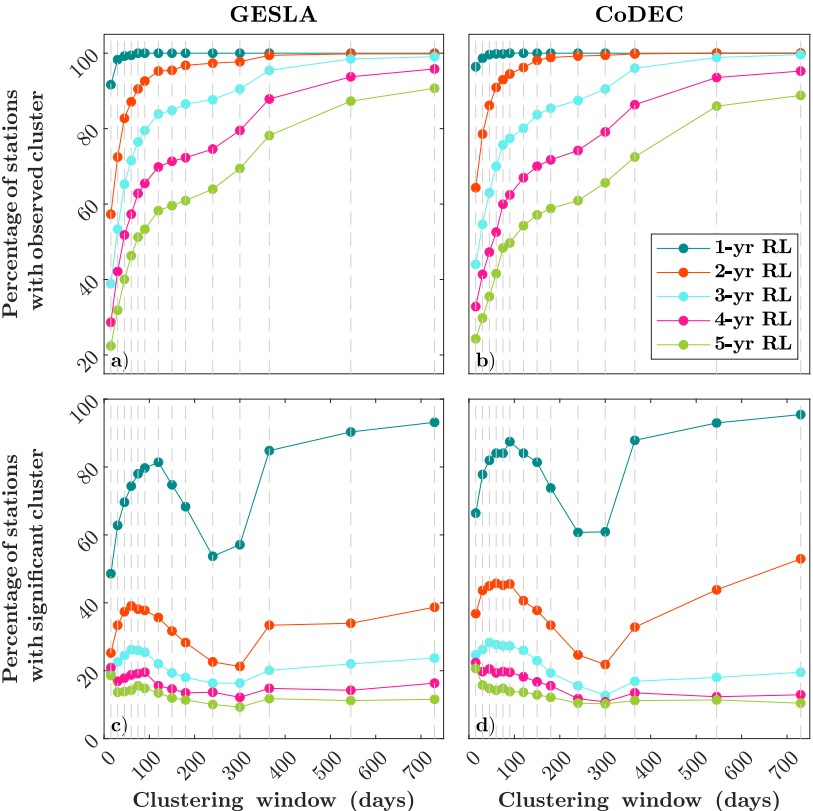

**Figure 2.** Identified clusters of storm surges at 527 coastal sites using varying thresholds and clustering windows. (a, b) Percentage of locations where clustering occurred based on GESLA (a) and CoDEC (b) data. (c, d) Percentage of locations where clustering behavior differs significantly from the assumption of a Poisson distribution for GESLA (c) and CoDEC (d) data.

in a way that aligns with a Poisson process. For example, at the 1-year return level, nearly all stations show clustering, and 70% (across all window definitions) exhibit statistically significant clustering. Meaning only ~30% of those time series exhibit clustering behavior that follows a Poisson distribution. For instance, at a specific station in Brest (France), we derive a $K$ value of 1.7 from the observations (for the 1-year return level and 365-day clustering window), exceeding the 98th percentile of the Poisson-based bootstrap distribution (equivalent to a $K$ value of 1.4), indicating statistically significant overdispersion ($P \approx 0.01$). Similar patterns are observed at 95% of stations, where the likelihood of such clustering occurring under a Poisson process is generally below 1%.

Moreover, our results indicate that the CoDEC data (Figure 2b,d) is overall able to reproduce the temporal clustering of storm surges observed in GESLA (Figure 2a,c). These differences are spatially explored below. The same results as shown in Figure 2b,d, but obtained using all the model points included in the CoDEC dataset, can be found in Supplementary Figure S4, showing overall similar behavior.

Next, we examine how temporal clusters vary from one location to another. Figure 3 shows the percentage of clustering, defined as the number of clusters divided by the total number of events, for the 1 and 3-year return levels, again for GESLA (Figure 3a,c) and CoDEC (Figure 3b,d). We use a 90-day clustering window definition based on the peaks of significant clustering in Figure 2c,d.

Higher clustering values, up to 70%, exist at the 1-year return level in regions like northern Europe and northern Australia. In other words, when a storm surge exceeds the given threshold in those regions, 70% of the time another event occurs within <90 days. A latitudinal difference is observed in Australia and Japan, with

equatorial regions showing higher clustering values. For the 1-year return level (Figure 3a,b), stations without significant clustering (no black contour) are found in northern Japan and parts of the US East Coast. For the 3-year return level (Figure 3c,d), stations without significant clusters are distributed more evenly across coastlines. Results may vary depending on the threshold or clustering definition (see Supplementary Figure S5). Considering longer windows leads to a higher percentage of clustering, particularly at the lowest threshold, whereas at the highest threshold (5-year return level), results are more consistent across different clustering window definitions.

Results from CoDEC (Figure 3b,d) resemble those obtained from GESLA (Figure 3a,c), with mean (median) differences (i.e., absolute difference in the percentage of clusters) of 10% (7.5%) for the 1-year and 12% (10%) for the 3-year return level, and standard deviations between stations of 8 and 11%, respectively. More than half of the stations, 62% for the 1 year and 52% for the 3 years, show differences (in the percentage of clusters) below 10%. Some differences persist in the Mediterranean Sea and some parts of the Baltic Sea. However, the largest discrepancies appear to be randomly distributed, suggesting they are likely due to site-specific characteristics rather than a systematic bias. Such characteristics can include influences of freshwater discharge, which would be captured by tide gauges but not in the model data, or differences in the predicted tides (which are calculated using harmonic analysis for the GESLA data and predicted by a hydrodynamic model in the case of the CoDEC data). However, on a global scale, these differences are minor, and the spatial clustering patterns are consistent across both datasets.

Finally, the same analysis was repeated using skew surge data (calculated for all 527 stations, both for tide gauges and CoDEC

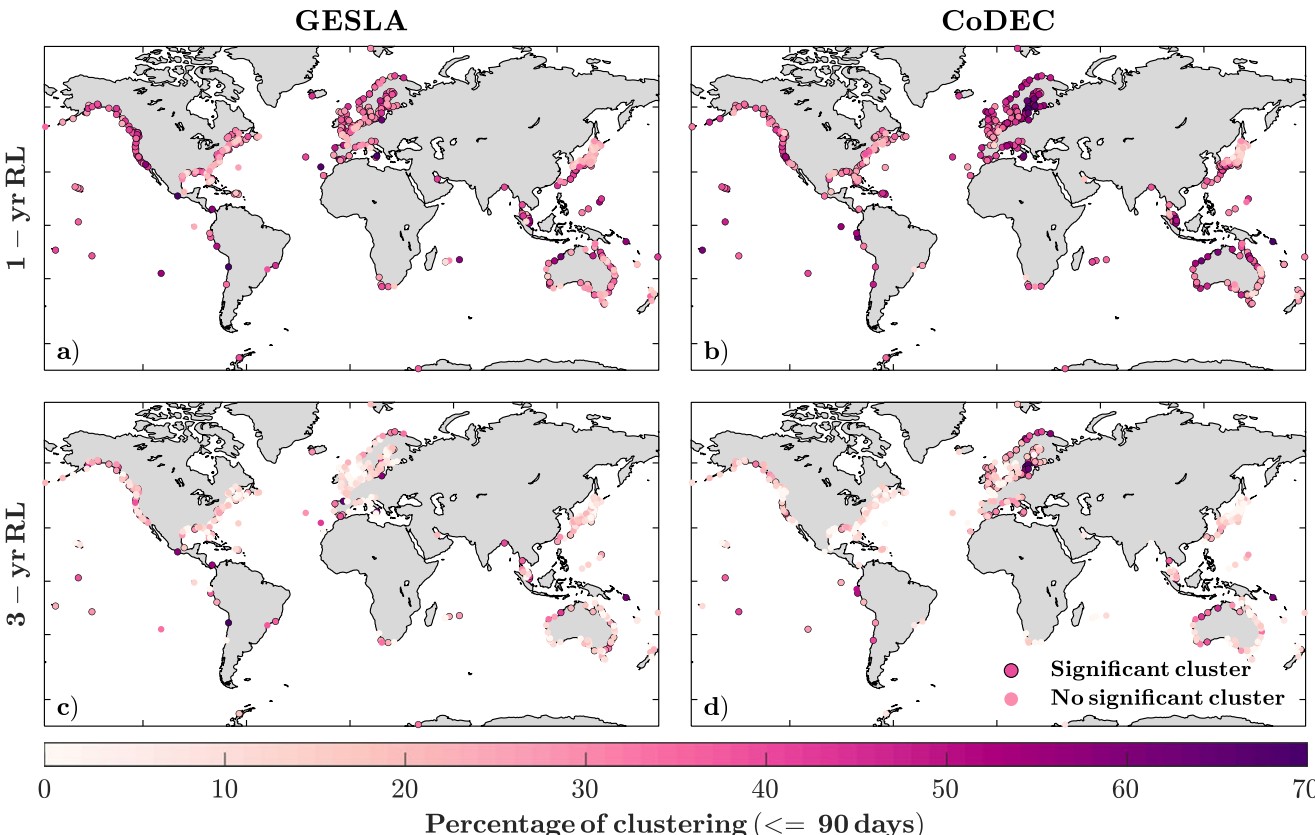

**Figure 3.** Percentage of clustering (i.e., number of clusters divided by the number of events) when using a 90-day clustering window and a 1- (a, b) and 3-year (c, d) thresholds to identify storm surge events. Results are shown for GESLA (a, c) and CoDEC (b, d) data.

time series), where skew surge represents the difference between an observed high water and the closest predicted high tide. This analysis is to ensure that our findings are robust against the storm surge definition and not substantially influenced by tide–surge interaction. Supplementary Figures S2 and S3 show that the results are consistent regardless of the definition.

### Global analysis

After validating CoDEC against observations, we identify the temporal clusters of storm surges at the global scale, using all 18,719 CoDEC coastal model points. To be consistent, we show the results from defining clusters as those events occurring within a 90-day time window; results for other cluster definitions (time windows and thresholds) are shown in Supplementary Figure S5. On average, 39% of the 1-year return level events globally were part of a cluster (Figure 4a). This value drops to roughly 10% when calculated using the bootstrap method explained in "Methods" section (consisting of 1,000 Poisson time series with the same rate of occurrence as the observations). Storm surges around Oceania, the Baltic Sea, and the Caribbean Sea exhibit a higher tendency to cluster, where, in some cases, up to 70% of the events were part of a cluster. Overall, these clusters (defined by <90 days and a 1-year return level threshold) contain an average of 2.5 events, with some regions, such as the Baltic Sea and parts of Indonesia, reaching up to 4 events on average. However, at most stations (~66%), clusters typically consist of just two events. Similar to what was shown before, fewer temporal clusters occur when higher thresholds are considered. On average, 14% of the 5-year return level events were part of a cluster, while the average value obtained using a Poisson

distribution would be 2.5%. As with the 1-year return level events, Oceania, the Baltic Sea, and the Caribbean Sea show a higher tendency for clustering with values up to 60% (Figure 4b). Overall, temporal clustering above the 1-year return level is significant at 92% of the model points, decreasing to 25% for the 5-year return level (see Supplementary Figure S6). Places with no significant clustering are mainly found in the northern part of Japan and the East coast of South America (mostly Argentina and Uruguay), where clustering values are below 20%.

So far, we have examined how clustering varies across different return levels. Next, we examine the composition of these clusters, specifically the contribution of different return level events to clusters defined as ≤90 days apart. Since a single cluster can consist of events with different intensities (e.g., one exceeding the 3-year return level and another the 5-year return level), analyzing this composition helps to better characterize the nature of clustered events. To quantify the contribution of different event intensities to clustering, we use the cumulative distribution function of the percentage of times different events with different return periods were part of a cluster (Figure 5a) versus how often such events occurred without being part of a cluster (Figure 5b). For example, when two events cluster (inter-arrival time ≤ 90 days), at 50% of the coastal points where that happened, at least 50% of the clustered events fall within the 1- to 2-year return level range (blue line in Figure 5a). The remaining contributions come from events reaching the 2- to 3-year return level (15%), 3- to 4-year return level (7%), 4- to 5-year return level (4%) and ≥5-year return level (21%). This distribution reflects the average contribution of events above certain thresholds to clustering.

As expected, the lower return level range (1–2 years) contributes the most to clustering. As return levels increase, the curves move to

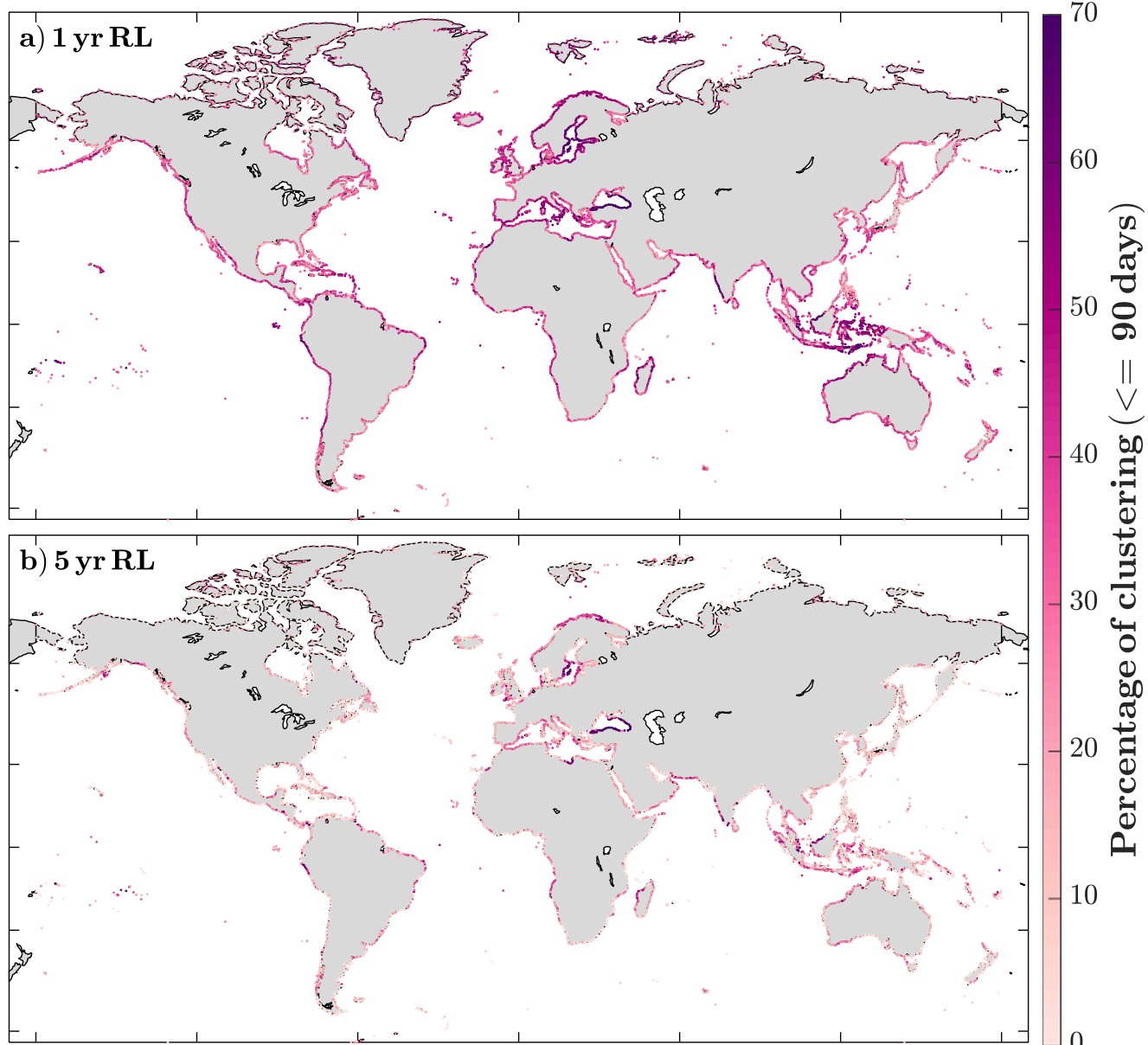

**Figure 4.** Percentage of clustering, defined as the number of clusters divided by the number of events, for clusters of <90 days. For events above the 1- (a) and 5-year (b) return levels, for all CoDEC coastal grid points.

the left, indicating decreasing contributions to clustering. The curve representing the 5-year return level is an exception, and this is because all events that reach or exceed that return level are included here, and not just the ones falling into a narrow band of two different return levels, as is the case for the other return levels analyzed here. Hence, this category shows a significant influence, ranking second in contribution after the 1- to 2-year return level range. Figure 5b presents the same analysis for nonclustered events (i.e., the number of times 1- to 2-year events, e.g., were not part of any cluster). The results closely resemble those in Figure 5a, indicating that there is no systematic trend in the relative contribution of events with certain intensities to clustering versus nonclustering.

As shown in Figure 5a, events exceeding a 5-year return level also contribute substantially to clustering. To further examine this, Figure 5c focuses on cases where at least one event within a cluster is at or above the 5-year return level. This analysis quantifies the percentage of times (for each grid point) that a ≥5-year return level event clusters with a smaller event (purple line) versus when it clusters with another event at or above the 5-year return level (green line). Results indicate that for at least 50% of the coastal points, events exceeding the 5-year return level always cluster with lower-threshold events. In the remaining 50% of the points, the contribution of high-intensity events, while not zero, remains low. On average, 87% of these clusters occur with a lower-threshold event, while only 13% involve another event at (or above) the 5-year return level.

### Inter-arrival time analysis

Examining the inter-arrival times between events allows us to provide a spatial representation of the average time between events globally. Also, as suggested by numerous studies (Dollar and

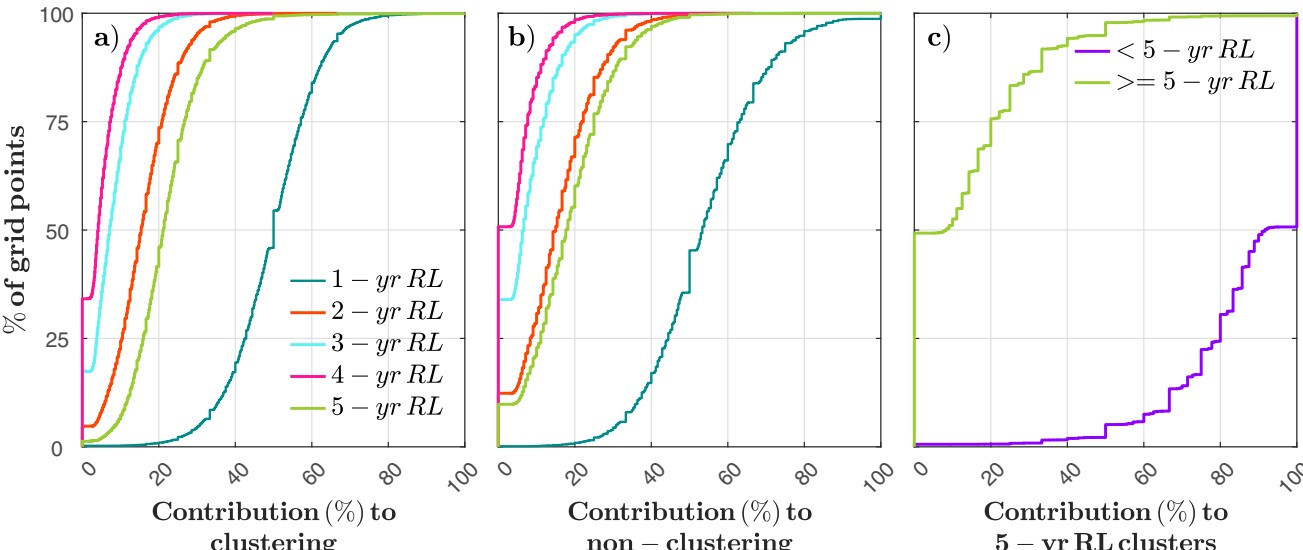

**Figure 5.** Cumulative distribution function of the contribution of different return levels to (a) clusters and (b) the contribution of events that are not part of clusters. The cluster definition used is ≤90 days, and colors indicate the intensity. Panel (c) shows the contribution of events below (purple) and above (green) a 5-year return level to cluster with another 5-year return level (or higher) within 90 days.

Tribble, 1993; Merz et al., 2016; Besio et al., 2017), the time between two consecutive clusters can be as important as the time between consecutive events; it depends on the type of impact one is concerned about. We can distinguish these two stages of clustering as short-term clustering and recovery time. Short-term clustering refers to periods where events occur in rapid succession, leaving little to no time for recovery between them. In contrast, recovery time represents the longer intervals between clusters rather than individual events. This phase provides a critical window for environments to regenerate, communities to rebuild and infrastructure to be restored before the next clustering period begins. We analyze both stages independently by splitting the inter-arrival time analysis into two. The stages are defined using 180 days as boundary (i.e., inter-arrival times ≤ 180 days become part of the short-term clustering, and we define recovery times as the periods between events when they are more than 180 days apart); the 180 days threshold was chosen based on a shift in the clustering behavior in Figure 2 around that value, which generally aligns with the typical duration of stormy seasons (e.g., North Atlantic hurricane season from June to November, or the stormy season in Europe from October to March). Figure 6 focuses on the short-term clustering, using the median (Figure 6a) and standard deviation (Figure 6b) of inter-arrival times shorter than 180 days, for events exceeding the 1-year return level. The shortest values occur around Europe and southern Asia, with localized minima in the Southern Adriatic Sea, Java (Indonesia) and northern Madagascar, where values fall below 1.5 days, commonly used as an independence window in storm surge studies. However, such short intervals represent <6% of the model points. Other notable regions, with short inter-arrival times, include the Caribbean islands and northern Australia, with values around 15 days and standard deviations in the order of 20 days. The average inter-arrival time across all coastal points is 26 days, with a standard deviation of 90 days. By contrast, estimating the average inter-arrival time using an exponential decay model with the same rate of occurrence (i.e., assuming a Poisson distribution) gives a value of 90 days, with minimal variation between stations.

Figure 7 examines the time periods without clusters (in years; Figure 7a) and the recovery times (inter-arrival times >180 days;

Figure 7b). Figure 7a shows the number of years without events, that is, inter-arrival times longer than >360 days (twice the threshold we use to separate short-term clusters); it is noteworthy that years do not need to be consecutive (i.e., a 360-day period with events can occur between two 360-day periods without events). Overall, values show little variability, averaging 13 years (± 2.5 years) without events, over the 40-year record. Only 17% of points exceed 15 years, and just 0.15% reach 20 years. The lowest values are found in the Black Sea, where values drop below 5 years. Figure 7b shows the median recovery time, which for 91% of the stations is <2 years. The average time between events is 1.5 years, with higher values where clustering occurs more often (Baltic Sea and the Sunda Islands). This indicates the time window that communities and natural ecosystems have, on average, to recover.

## Discussion

This study presents a global analysis of the temporal clustering of storm surges. We identified the temporal clusters using tide gauge observations, which are spatially sparse and discontinuous but provide accurate in situ sea-level information. We also use a global reanalysis dataset, which allows us to identify the temporal clusters of storm surges everywhere along the global coasts.

The CoDEC model validation shows strong agreement with tide gauges from GESLA. We hypothesize that discrepancies arise primarily in regions where external factors, such as freshwater discharge or tidal differences, influence surge behavior. Additionally, uncertainties may stem from the declustering process or limitations in model-based time series, but these remain within reasonable margins for a global-scale study.

The validation was also extended to different definitions of clustering, which can significantly influence the results. The selection criteria for specific clustering definitions are largely dependent on the local characteristics of the place of study (de Ruiter et al., 2020); therefore, given the global scope of this analysis, selecting a unique fixed window was not appropriate. Instead, we selected a range of clustering windows supported by literature and assessed how

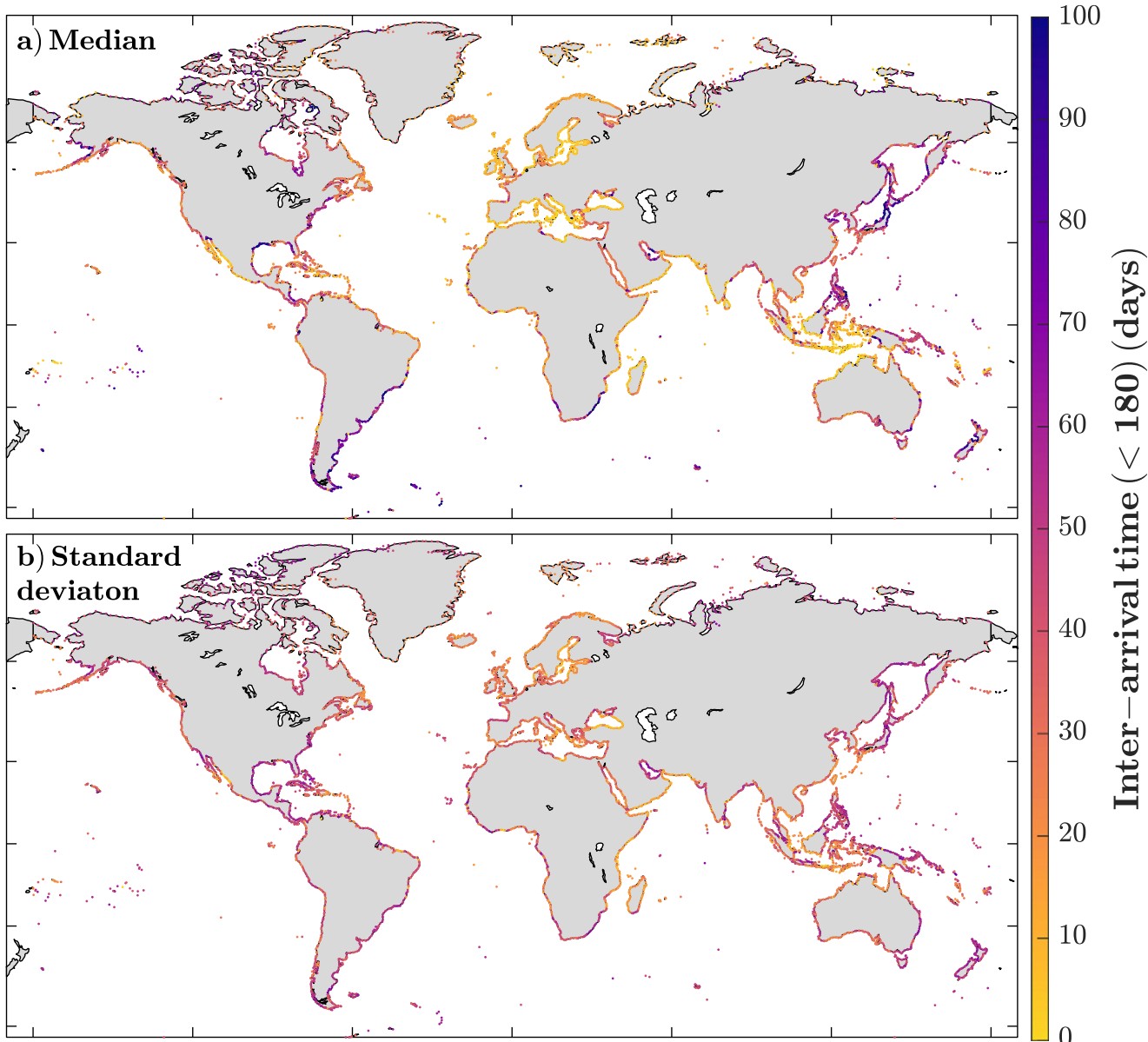

**Figure 6.** Median (a) and standard deviation (b) of the inter-arrival times shorter than 180 days for all CoDEC coastal grid points and events above the 1-year return level.

clustering behavior varies across different definitions (following other studies such as Santos et al., 2017; Jenkins et al., 2022; Brunner and Stahl, 2023). Moreover, the results depend on the type of events considered. While extreme events have been extensively studied due to their high impact, moderate events are often overlooked. However, they play a key role in clustering dynamics, as their short recurrence intervals amplify cumulative impacts. The inclusion of these events is essential for a comprehensive understanding of storm surge clustering and its implications (Towe et al., 2020). Accordingly, our analysis focuses on a range of low to moderate return levels (1- to 5-year return levels) while still accounting for the most extreme events, as those exceeding the 5-year return levels are also included. Results indicate (Figure 2) that the cluster frequency decreases as event intensity increases, with clustering behavior strongly influenced by the time window used for defining clusters.

Storm surge event frequency has traditionally been modeled using the Poisson distribution. However, the presence of clustering challenges the suitability of this assumption. To assess the Poisson assumption, we compare the observed clustering (Figure 2a,b) to the clustering expected under a Poisson process with the same occurrence rate (Figure 2c,d). While the Poisson distribution performs better at higher thresholds, it fails to adequately represent clustering in at least 20% of stations, regardless of the chosen time window. This discrepancy increases to over 80% as the threshold decreases and more events are included.

Additionally, we observe a sharp decline in significant clustering between 120 days and 1 year, suggesting a shift in clustering behavior. Beyond 1 year, the clustering stabilizes; this could be linked to the fact that when considering a 1-year window (or longer), we capture two or more storm seasons in certain areas, resulting in a more consistent number of events within the clustering window. Shorter clustering windows exhibit greater variability, with peaks in significant clustering around 90–120 days, indicating that clustering dynamics are highly sensitive to the chosen definition. Overall, the results highlight the need for alternative approaches to model event frequencies, particularly when moderate events are considered. These events

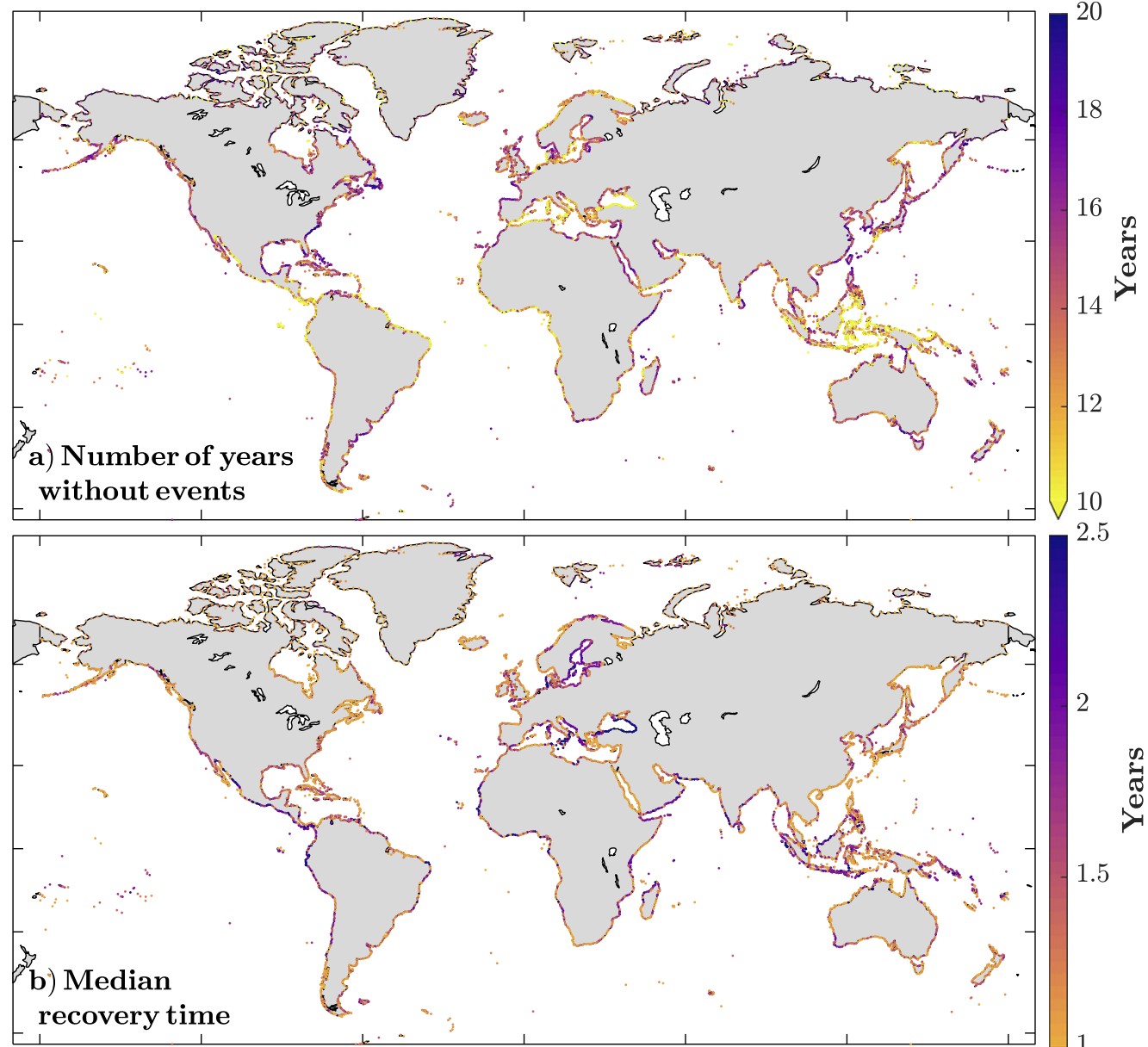

**Figure 7.** Number of years without any events above the 1-year return level (a) and median time between events when they occur >180 days apart from each other (which we define as recovery time).

can be as impactful as extreme ones when the recovery time between them is insufficient (Karunarathna et al., 2014).

Importantly, while the 1-year return level defines the minimum event intensity included, clusters can involve higher-threshold events, which can create greater damage. We quantify the contribution of different return levels to these clusters (Figure 5). Lower-threshold events (1-year return level) show the highest contribution to clustering, and contributions decline while increasing the return levels, except for the 5-year return level, which encompasses all higher events and thus increases again. These results indicate that extreme events make a considerable contribution to clustering. While this study primarily focuses on low to moderate return levels, the clustering of more extreme events remains crucial. We perform a detailed assessment of high-threshold clustering, imposing an additional criterion requiring at least one event at or above the 5-year return level. The contribution is further divided into events

above or below the 5-year return level that cluster with another event exceeding this threshold. Our results indicate that ~63% of the stations experienced two events (at or above a 5-year return level) within 90 days at least once over the study period (i.e., 63% of the stations experience at least one cluster formed by two events at or above a 5-year return level). On those stations, on average, this is the composition of 22% of the clusters (<90 days). Overall, the lowest return levels contribute the most to clustering, with very little contribution from other high return level events clustering together (within 90 days).

The widespread presence of clustering across nearly all stations suggests that assuming the frequency of storm surge events follows a Poisson distribution may not be appropriate. Instead, alternative models should be considered to account for clustering. For instance, Jagger and Elsner (2012) proposed a statistical model based on the Poisson distribution with an extra parameter to account for clustering.

The main idea is that while events can cluster, each cluster remains independent of the others and follows a Poisson distribution. The statistical implications are the existence of two important "windows" in the event frequency distribution: (1) the clustering size or short-term clustering, and (2) the recovery time, which follows an exponential decay.

We present a quantitative approach to this concept, studying the inter-arrival time distributions. Short-term clustering (≤180 days, Figure 6) reveals high event recurrence, with a median inter-arrival time of 26 days and notable regional variability (standard deviation: 90 days). The lowest values are found in Europe, the Philippines, and the Caribbean islands. Clustering implies, among other things, the existence of calm periods in between. This recovery time has been extensively studied, compared to short-term clusters, particularly in natural environments, due to its critical role in the ecosystem growth and recovery from extreme events (Mumby et al., 2011; Karunarathna et al., 2014; Yao et al., 2022). For these longer inter-arrival times (>180 days, Figure 7), we find that, on average, CoDEC stations experience calm periods of ~1.5 years between clusters. The fact that the average recovery time is longer than a year means there are some years that do not experience any event; on average, those represent 32% of the study period (i.e., 13 out of 40 years exhibiting no events).

While our study focuses on identifying where storm surge clustering occurs, understanding the underlying mechanism of why this happens remains an open question. Most studies aiming to understand the mechanisms behind storm clustering have focused on cyclone genesis and track behavior. On daily timescales, clustering is often linked to the development of cyclone families or secondary cyclogenesis, described by Bjerknes and Solberg, (1922). In these cases, storms are not independent; rather, secondary cyclones form along the frontal boundaries of preexisting primary cyclones, resulting in successive storm events. However, not all storm clusters can be explained by the cyclone family mechanism. As noted by Dacre and Pinto (2020), storms may also cluster purely by chance. Even when cyclone occurrences follow a random process (e.g., Poisson), some clusters are statistically expected to occur.

On longer timescales, storm clustering can be modulated by large-scale modes of climate variability. Many studies have used these teleconnection patterns to estimate the likelihood of active or inactive storm seasons, often using regression models (e.g., see Mailier et al., 2006; Vitolo et al., 2009; Economou et al., 2015). However, despite growing interest, much of our current understanding of serial cyclone clustering remains centered on the North Atlantic and European sectors (Dacre and Pinto, 2020; Xi and Lin, 2021; Xi et al., 2023).

More recently, research has begun to explore storm clustering from a storm surge perspective, linking successive storm surge events to large-scale ocean–atmosphere interactions (Jenkins et al. 2022). In the United Kingdom, several studies have associated consecutive storms with the North Atlantic Oscillation (NAO) and the West European Pressure Anomaly (e.g., see Woodworth et al., 2007; Castelle et al., 2017; Santos et al., 2017). Other regions also show linkages between storm clustering and climate drivers. For instance, clustering near Florida has been connected to the NAO and the Southern Oscillation Index (e.g., see Jagger and Elsner, 2012), while in Madagascar, storm activity has been linked to El Niño Southern Oscillation and the Madden–Julian Oscillation (e.g., see Nash et al., 2015; Macron et al., 2016). Nonetheless, the interaction between synoptic-scale processes and large-scale climate modes remains an open area of research, requiring further investigation across different regions and storm types.

Finally, despite ensuring the highest possible accuracy in our analysis, certain limitations must be acknowledged. The use of observational data introduces challenges, such as outliers, gaps and inconsistencies, which were addressed as much as possible through extensive preprocessing. Similarly, model-derived data can present validation issues, particularly in regions lacking in situ observations. However, our validation efforts focus not on the full validation of the CoDEC model itself, but specifically on its ability to reproduce clustering behavior. Another source of discrepancies are the return level estimations. The methodology used to estimate return levels is consistent across datasets; however, some differences may result from the declustering procedure. However, the declustering method has been previously validated and shown to be robust and applicable to both tide gauge time series and CoDEC data. Minor discrepancies may also stem from magnitude biases in CoDEC, although a general inspection indicates that the model captures event magnitudes reasonably well (here, the exact magnitude is less relevant as long as peaks exceed our thresholds of interest at the same time the tide gauges recorded a storm surge event). Finally, during the validation, the presence of gaps in the observational records (which were matched in CoDEC for consistency) may result in missing events and slightly lower cluster counts (in both datasets). However, this issue is mitigated in the global analysis, where CoDEC's complete and continuous time series are used. All the methodological choices made were to provide a reliable framework for assessing storm surge clustering globally, highlighting the need for regional-scale analyses to further understand the mechanisms driving storm surge clustering and its implementation.

## Conclusion

Storm clustering remains an underrepresented factor in coastal flood risk assessments despite its significant impact on coastal communities and ecosystems. How communities and natural ecosystems are prepared for back-to-back events will determine their ability to recover. Properly understanding the probability of these consecutive events and better characterizing their frequency is the first step to enhancing the resiliency of coastal communities to repeated shocks.

While storm clustering is inherently site-specific, this study provides a global perspective by analyzing different clustering definitions, event intensities and their spatial distributions. Moreover, we evaluate the applicability of the commonly used Poisson distribution and show that for relatively short return periods, this assumption is not satisfied at the vast majority of locations. Finally, we examine two important clustering definitions, short- and long-term clustering, which provide useful information about the recovery time between storms for each station. On average, the recovery time is limited to <2 years for 90% of the sites. These results underscore the importance of considering moderate events in clustering analyses, challenging the adequacy of Poisson-based assumptions and highlighting the need for alternative frequency models that account for event dependencies. Such models could then be used, among others, to generate boundary conditions for coastal impact models (such as the Dynamic Interactive Vulnerability Assessment model (Hinkel and Klein, 2009) and the Coastal Impact and Adaptation Model (Diaz, 2016), which currently lack this information. This also includes integrating storm surge data with tidal information, an aspect not explicitly addressed in our study but essential for advancing toward comprehensive hazard and risk assessments. Finally, the adaptability of this framework provides a useful basis for extending the analysis to other types of

hazards, enabling the inclusion of spatiotemporal patterns of consecutive events in broader risk assessments.

**Open peer review.** To view the open peer review materials for this article, please visit http://doi.org/10.1017/cft.2025.10008.

**Supplementary material.** The supplementary material for this article can be found at http://doi.org/10.1017/cft.2025.10008.

**Data availability statement.** All the data are publicly available. CoDEC data are available at the Climate Data Store (CDS) of the Copernicus Climate Change Service (C3S) at the following url: https://cds.climate.copernicus.eu/. GESLA-3 dataset can be downloaded at https://gesla.org/. The data used in the study will be available and can be downloaded from a Zenodo data repository (10.5281/zenodo.15097911). We provide the declustered time series for the 18,719 CoDEC coastal points for the 5 return levels used in the study and the list of the tide gauges used for validation.

**Acknowledgments.** A.R.E. acknowledges support by the European Union Horizon 2020 EXCELLENT SCIENCE – Marie Skłodowska-Curie Actions (Grant number: 101019470).

**Author contribution.** Funding acquisition: T.W. Data curation: A.M. Formal analysis: A.M. Methodology: A.M. and R.J. Project administration: T.W. Supervision: T.W. Writing original draft: A.M. and A.R.E. All authors approved the final submitted draft.

**Financial support.** This work was supported by the United States National Science Foundation (Grant number: 2141461).

**Competing interests.** The authors declare none.

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
