## [Reviewer Report]

The paper, ‘Global Analysis of Temporal Clusters of Storm Surges’ by Martin et al. assesses the likelihood that extreme storm surge occur in quick succession using both global tide gauge and model datasets. Overall, I think the idea of the paper is novel and an important research topic, however several revisions are required to strengthen and justify the findings.

Main comments

Focus/scope of paper: The abstract makes it appear as if testing whether it is Poisson distributed is the main objective, whereas the title and most of the manuscript makes it sound more general: to study clustering of storm surge events and the implications of this for hazard modelling. I think the paper as it currently stands would be stronger (and require fewer revisions) if it is the latter, so I am going to base my subsequent feedback on this.

Storm surge terminology: e.g., “Among the spectrum of coastal hazards, storm surges…” Ultimately the hazard is flooding or erosion (refer next comment about erosion). At the very least you need to replace references to “storm surges” to be “extreme storm surges” or “extreme positive storm surges” as the way this paper (and most others) define storm surge they have mean zero and can be positive or negative. By only considering storm surges with return periods of between 1 and 5 years this paper only considers large positive storm surges. I think it is beyond the scope of the paper to consider in detail, but it could be worth mentioning how the ability for storm surges to cause flooding are modulated by tides and this is a limitation of the study’s relevance for flood risk assessments (e.g., Enriquez et al. 2022: https://doi.org/10.1029/2021JC018157, Hague et al. 2023: https://doi.org/10.1029/2023EF003784, Palmer et al. 2024: https://doi.org/10.1029/2023JC020737 ).

Erosion and scope of paper: e.g., “moderate storms can resemble the erosion caused by a single extreme event” and “ Storm surge events impact a wide range of natural ecosystems, including beaches, dunes, as well as public infrastructure and coastal communities”. The data used in this paper cannot be used to infer erosion hazards as GESLA and CODEC do not include the impacts of wave setup or runup on coastal water levels. To resolve this, discussion of erosion and open coast (e.g., beaches) impacts need to be removed, or alternative data sources that include waves need to be considered. Ideally alternative data sources would be the solution as I suspect cumulative impacts of erosion are in important part of the impacts of clustering, but I understand this is a lot of additional work so it is sufficient to limit the scope to still water level or storm-tide.

Tide-surge independence: Ultimately the delineation of what part of the water level is considered “surge” and what is considered “tide” is subjective (e.g., see Hague & Taylor 2021: https://doi.org/10.1007/s11069-021-04600-4). This follows from tides being defined via harmonic analysis which is fitting sums of sines to observations as a statistical (i.e., non-physical) approximation of (un-observable) physical processes (more on this later). As storm surges are defined as “not tide”, this means that storm surge is also impacted by how tides are defined. There has been a lot of work done on the interactions between storm surges and tides and other water level components (e.g., Arns et al. 2020: https://doi.org/10.1038/s41467-020-15752-5, Horsburgh & Wilson 2007: https://doi.org/10.1029/2006JC004033, Moftakhari et al. 2024: https://doi.org/10.1029/2023AV000996, Idier et al. 2019: https://doi.org/10.1007/s10712-019-09549-5). This has led to the promotion of skew surges as an alternative to the non-tidal residual because it is largely independent of tides – see Williams et al. 2016 https://doi.org/10.1002/2016GL069522. It would be worth repeating at least part of the analysis using a skew surge metric to assess the sensitivity of the results to different metrics for the physical process of extreme storm surges. This could verify that the findings are not simply due to the choice to use the non-tidal residuals as the metric for storm surge – the distribution of which is likely modified by tide-surge interactions. This could be something like what is described in the first paragraph of Section 2.2 – I found this approach to sensitivity testing very good!

Year-by-year analysis: Performing harmonic analysis on a year-by-year basis without SA and SSA means that seasonal variations in tidal heights due to the seasonal cycle of mean sea level are included as part of the “storm surge” component of the water level rather than the tides. There isn’t anything wrong with this per se but to facilitate fair comparisons with CODEC, it would be best to handle the seasonality in a consistent way for both gauge and model estimates of storm surges. Please include some discussion on this.

More details on comparison between GESLA and CODEC: I liked sentence: “ However, on a global scale, these differences are minor, and the spatial clustering patterns are consistent across both datasets.” It would be great to see this statement justified with some data (e.g., RMSE or percentage error) to quantify that the differences are “minor”.

Context for statistics: I struggle to interpret the statistic “14% of the 5-year return level events were part of a cluster” and others in Section 3.2. Am I meant to be impressed by how high, low or expected that number is? It says in methods that Ripley’s K metric is used. Can an assessment using Ripley’s K be provided to give an indication of the notability (and sign e.g., higher or lower than expected) of such results? Similarly, for “the average interarrival time across all coastal points is 26 days” in Section 3.3.

Riverine discharge in GESLA: The discussion includes: “We hypothesize that discrepancies arise primarily in regions where external factors, such as freshwater discharge” This could be tested using the GESLA dataset which classifies sites as “coastal”, “river” or “lake”. For example, does CODEC perform worse when compared “river” gauges than “coastal” gauges?

Implications for coastal hazard assessment: The abstract states that “These findings offer insights into temporal clustering dynamics of storm surges and their implications for coastal hazard assessments.” More details are required on the implications for coastal hazard assessments. For example, what do coastal hazard assessment methods assume that is invalidated by the findings? How different are results of risk assessments if clustering is considered? It could be good to link the coastal adaptation literature which often details how risk assessment information is used (e.g., Haasnoot et al. 2024: https://doi.org/10.1016/j.gloenvcha.2024.102907).

Minor comments:

Last Paragraph Page 11: “randomly distributed” – specify the distribution as lots of possible ways for random variables to be distributed.

Last Paragraph Page 10: I don’t the repetition of the methods is required here, could rephrase as “…challenges the suitability of this assumption. We compare the observed clustered…”

---

## [Reviewer Report]

Summary

This paper examines the temporal distribution of storm surge events, an important topic and one that deserves greater attention. Overall, this paper presents compelling evidence of spatial and temporal patterns in event timing which indicate that commonly adopted methods (e.g., random distribution) may be unsuitable for coastal planning needs. The clustering and timing of events is critical for coastal hazard assessment because the amount of time between arrival times of individual storm events may be more consequential than the magnitude of any one of those events. This is important both for shorter than average inter-arrival times, where the impact of an event compounds the impacts of a recent event (or events), and for longer than average inter-arrival times, where greater potential of recovery increases the resilience to future events. I congratulate the authors on the in-depth and comprehensive approach to examining clustering and timing and the global application of the analyses. There are some questions remaining that I believe should be addressed to better explain the results and thus make this work more impactful.

The five major points which should be addressed are as follows:

1. While the justification for the work is well articulated, the key themes and conclusions would benefit from clearer structuring in the introduction, and signalling (and ordering) the distinct steps throughout the methods, results, and discussion – e.g., by including a paragraph at the end of the introduction that explains how the paper will proceed, and by structuring the ensuing methods/results/discussion stepwise with sub-headings.

2. Figure 1 is great for illustrating the importance of clustering, it would be helpful to provide an additional figure in the methods section clearly illustrating the different thresholds and windows (perhaps using the same Apalachicola tide gauge). This would help in interpreting the pattern in the identification of significant clusters presented in Figure 2. The relevance of seasonality could also be highlighted in Figure 2 (e.g., by shading the length of time feasibly defining an overlap in two storm seasons ~400 days). The effects of different RLs and cluster windows on the identification (and definition) of clustering is key.

3. The importance of the topic is well stated, but the importance of the contribution of this work needs to be clearer. There are a number of advances in methodology that are not prominent enough, and some missed opportunities to present a useful approach for application. To present clearer evidence of the contribution and for a stronger conclusion, the strengths and limitations of each preceding step need to be clearly stated and quantified well before the final paragraph. Specifically:

a. CODEC model validation

b. Return level estimates

c. Clustering definition

The statement of limitations in the final paragraph of the discussion skipped over these points, missing the level of detail required to be convincing.

4. The definition of clustering should be more precisely stated following the analysis of clustering thresholds and inter-arrival times, and the effects of different choices discussed. What is the threshold and clustering window of greatest use for coastal planning? Should 1-YR RL events be included in the definition of a cluster (isn’t recurrence within 1-year consistent with the assumption of random distribution)? Is a 730-day window justified/supported in literature for identifying clusters? Is it logical to say that including more frequently occurring events results in higher clustering?

5. From the results onwards, the authors discuss the presence or significance of clustering defined as two events within the specified time window. It would be highly relevant to further quantify and discuss the average and variation in the number of events in identified clusters, their composition, and the possible role of non-stationarity in spatial variations – especially as the GESLA observations are compared over different decades. One important point that I felt was lacking was the role of dependency within clusters, in addition to between individual clusters, and the correlation between storm-dense seasons and external factors (which could likely be drawn from the literature rather than by extending the analysis).

In addition, I offer the following minor points for the authors to consider at their discretion:

• P1 line 42: the sentence “Here, we raise the question…” seems to contradict the second sentence – Poisson distributions are used to assess storm surge frequency (not clustering).

• P2 line 55: it isn’t clear to me why tropical cyclones occurring during certain seasons are an example of less detrimental long-term effects.

• P3 line 29: this is a good point, but I didn’t see how the preceding sentences and references justified why the definition of a cluster is dependent on local and regional characteristics.

• P4 line 21: more detail is required to describe the tidal analysis to ensure repeatability (i.e., default constituents? SNR?). Also, check typo in last sentence of this paragraph.

• P4 line 53: this paragraph could be rearranged/revised to improve clarity.

• P5 line 19: the K metric could be better utilised in the results and discussion (including a plot in the supp?) to justify your statements that the presence of clustering violates the Poisson distribution assumptions. A plot of K vs clustering window would be helpful following paragraph on P6 lines 15-23.

• P6 lines 15-17: this sentence didn’t make sense to me, how can the clustering violate the Poisson principal but also occur in a way that aligns with a Poisson process? This is the point in the manuscript where I began to get muddled in the cluster windows and RL thresholds used to define clusters in events.

• P6 line 52: this is interesting, I would also like to know what % of the time a 3rd and 4th event another event occurs (with a figure!).

• P7 line 19: please quantify “these differences are minor” and preferably include a map of the differences or RMSE.

• Section 3.2: I suggest avoiding the tendency to lead with explaining figures, rather than leading with analysis supported by figures.

• P8 line 41: “events exceeding high return levels can significantly contribute to clustering” could be better phrased.

• P9 line 6: “an alternative approach” reads like introducing a new idea, instead link to the introduction (justification).

• P9 line 28-29: the choice of 180 days could benefit from further (and quantified) justification.

• P9 line 54: “note that years do not need to be consecutive” came as a surprise here, would benefit from further justification.

• P10 lines 24-30: these statements seem a little vague, would be stronger supported with statistics.

• P10 lines 33-52: the first sentence seems to be self-explanatory, a more specific intro sentence would help to make the importance of this paragraph obvious. In particular the point regarding the relationship between cluster frequency and event intensity seems understated. Here is where I would have liked to hear what the preferred selection criteria are for defining clustered events.

• P11 lines 23-27: it would be beneficial to further tie your findings back to the literature… how do your results inform these alternative approaches?

• P12 lines 13-15: no need to redefine recovery time here.

• P12 lines 23-27: please clarify these results, for how many sites? (from GESLA observations or from CODEC?)

• P12 lines 29-39: the concluding sentence of the discussion relies on a little too much faith from the reader, your expansive analyses deserve more decisive recommendations.

• P12 lines 54-60: this presentation of the synthesis of results comes too late in my view.

---

## [Editor Report]

We have now received two reviews of your paper, and I am pleased to see that both reviewers were relatively happy with the paper. Their comments are mainly around clarity and structure, with some recommendations in relation to analysis/changes that could easily satisfy their concerns. You have not addressed the important role that tides play in modulating the effect of a surge, and the reviewer provides suggestions on how you can do this. In summary, I believe you should be able to address the reviewers' concerns adequately.

---

## [Reviewer Report]

The author/s has responded comprehensively to reviewer comments and I congratulate them on a really interesting and important research contribution. I have no hesitation in recommending this submission for prompt publication.

---

## [Editor Report]

Both reviewers are happy with your responses, and I am looking forward to seeing this work published.